# Knowledge, Decisions, and Norms: A Framework for Studying the Structuration of Spreadsheets in Social Organizations

**Patrick Allen Rose** [1,*] and **Anthony J. Pennings** [2]

1   Seoul Business School, Seoul School of Integrated Sciences and Technologies, Seoul 03767, Korea
2   Department of Technology and Society, The State University of New York, Korea, Incheon 21985, Korea; anthony.pennings@sunykorea.ac.kr
*   Correspondence: parose@assist.ac.kr; Tel.: +82-10-2971-2752

**Abstract:** This article applies Anthony Giddens' theory of structuration to analyze how digital spreadsheet technologies both produce and are the products of structures that reinforce and transform the institutionalized routine practices of workers in social organizations from which the effects of power flow. Specifically, it identifies the built-in capabilities and features of spreadsheet applications that are reconfigured to embed organizational structures within them. A framework is proposed to explain the ways spreadsheets are assembled to embody three general forms of modalities central to Structuration Theory: (1) interpretive schemes, (2) facilities, and (3) norms. The proposed framework characterizes specific spreadsheet properties by their roles in enabling how these structural modalities construct realities with digital information that predetermine organizational thinking and doing. Illustrations of spreadsheets-in-practice are given as evidence of how digital spreadsheets reinforce and change organizational structures through their widespread diffusion and use.

**Keywords:** Anthony Giddens; spreadsheets; information systems; structuration theory; modalities; norms; interpretive schemes; facilities; power relations; discourse





## 1. Introduction

Digital spreadsheets are constitutive technologies that have emerged in modern organizations to construct knowledge and shape human agency. A community of investigators has been using Anthony Giddens' [1] structuration theory to analyze information systems. A review of the literature uncovers hundreds of published articles that use structuration theory as an analytical tool or 'sensitizing device' in information systems studies [2]. Orlikowski, most notably, and also Jones, Yates et al., and Rose, Lindgren, Henfridsson, and Pfeifer et al., were the first to propose frameworks and methods for how Giddens' theories might be practically extended to examine more complex and contemporary information systems [3–8].

A central theme of the previously mentioned information systems studies is the notion that designers (software engineers, managers, accountants, etc.) embed them with structures (resources, values, expectations, rules, strategies, norms, traditions, culture, etc.) to enact management strategies that have the effect of conducting the daily routines of workers [2]. As Orlikowski explains: "human agents build into technology certain . . . rules that define the organizationally sanctioned way of executing that work" [3]. It is through embedded structures in spreadsheets, and other technologies that mediate human interactions, that knowledge produces the conditions for the legitimated actions and decisions of workers. In addition, Orlikowski argues that Structuration Theory identifies moral sanctions for actions judged illegitimate. Thus, information systems can operate to narrow possible choices and behaviors to those that are sanctioned or privileged within a social organizational context.

Giddens' Structuration Theory (Figure 1) provides a model for understanding organizations and social life. In this framework, he introduces modalities that manifest as

technologies which automate the interactions between structures and knowledgeable human agents. Modalities produce the effects of power by pushing or steering agents toward certain patterns of sanctioned routine behaviors that are aligned with an organization's interests [1]. Even though structural modalities attempt to provide the 'rules' for organizationally sanctioned actions and behaviors, agents draw on their own situated experiences gathered over time via memory, social cues, and signified regulations to inform themselves about what is an 'correct' action. They anticipate the consequences of their actions by considering the information, expectations, and potential outcomes. They learn to work within the guidelines of the organization to do the jobs they are assigned and how to read the micropolitical dynamics, thus 'negotiating' their situation.

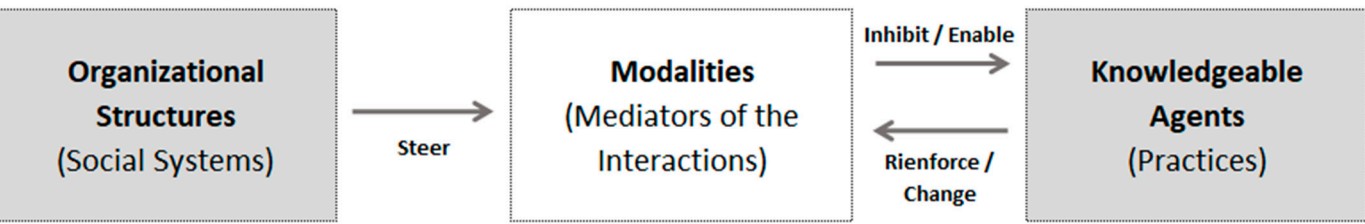

**Figure 1.** Structuration Theory Interaction Model. Adapted from Giddens [1].

Human agency is a central theme in structuration theory. Giddens believes that individuals have tacit and discursive knowledge of their situations but with bounded or limited rationality challenging their ability to truly act autonomously [1]. Autonomous knowledgeable human agents embed structures into information technologies that enable and limit their actions as technology operates across space and time. Giddens contends that agents can create and alter structures notwithstanding the structuring or mediation of interaction by value-laden technology. This notion of reflexive monitoring of human action is key to how structuration theory avoids the misconception that technology can be overly deterministic or controlling by stressing the central role of the recursive interaction between people and structures. Giddens conceives of this phenomenon as the 'duality of structure' in which people both purposefully produce and are themselves the products of structure and the power the exercise [1] (p. 191).

Structuration theory is intended to help understand how organizations are constituted and their transformative capacity to change. However, it does not provide a specific, apparent practical model for information systems research. Rose explains that "Structuration theory is too complex, diverse and alien to be adapted wholesale" [4]. The present article adopts the central concepts included in Giddens' theory of the duality and dimensions of structure (Figure 2) to use as a general tool for analysis. The previously mentioned three structural modalities are differentiated by their roles: (1) interpretive schemes to create meaning that mediates the processes of signification and reasoning, (2) facilities for exercising authoritative power or decisions over available resources, and (3) norms for invoking social rules and judgment to legitimate behaviors [3]. The model illustrates how power stems from the duality structure and agency which is recursively coordinated (represented by double arrows) in social systems across time and space by information technologies that allow agents to allocate resources to exercise power [1] (p. 125). Spreadsheet technologies emerged as an essential tool for collecting, categorizing, and systematizing the flows of data that aid the efficient management of an organization's authoritative and allocative resources.

Giddens argued that information technologies help social systems achieve "time-space power" [1] (p. 377). Electronic technologies store information over time and transmit it spatially. They distribute the arrangement of human activities, which determines how social relationships replicate, are transformed, and become stable. They eventually become the routine day-to-day, taken-for-granted practices in which actors habitually engage across the different settings and geographic distances [7]. Giddens uses the term social system integration to refer to the patterns of relationships and social reciprocity between

agents that are physically situated in different locations [9]. The present article shows how spreadsheets, embedded with structural modalities, are mediators of these cohesive effects.

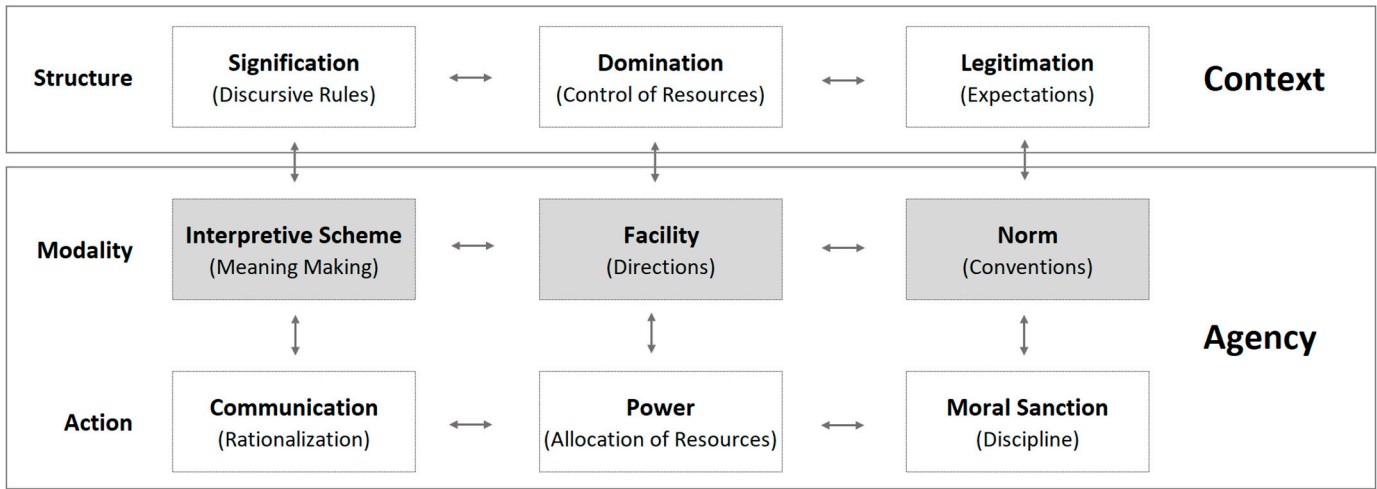

**Figure 2.** Structuration Theory—interactions between structures, modalities and actions. Adapted from Giddens [1].

Pfeiffer et al. review and compare several interpretations of Structuration Theory (e.g., Bright, Rogers, Burgelman et al., Jantsch, Orlikowski, Schwabe and Krcmar, and Cooper and Zmud) focused on describing how "technologies have an impact on organizations, their behavior and structure, [and] in turn, organizations and humans influence the use, the meaning, and further advances of technologies" [8]. They point out that most prior research regarding Structuration Theory and technology mainly examines the external relations between people and technology and does not consider the concept of the inner structure of the technology [8]. Orlikowski, for example, gives two definitions of information technology of on the one hand being malleable yet discernible entities with material properties and on the other hand being virtual settings or spaces for the constituting the prospects of interactions [3].

Synthesizing the perspectives discussed above, a conceptual model is proposed to apply Giddens' theories of structure, human agency, and time and space (Figure 3). For Giddens, "An organization is a collectivity in which the knowledge about the conditions of system reproduction is reflexively used to influence, shape, or modify that system reproduction" [1] (p. 13). The present article contends that spreadsheet technologies mediate the dynamic interactions between structures and knowledgeable human agents within modern organizations across time and space. They both produce and are the products of structures that reinforce and transform the generalized institutional practices of workers. The model attempts to operationalize Giddens' meta-level theories as a string of progressive layers: structures in digital spreadsheets that are historically embedded, constructed by knowledgeable actors, distributed through time and space via information systems and networks, mediated through widespread repeated use of the spreadsheets as social practices, and finally the spreadsheets transform over time as agents interpret the outcomes (reflexive monitoring) and communicate feedback back to the designers and managers about how to change the technology.

To summarize the literature review, studies that employ structuration theory are generally concerned with the dimensions of the duality of structure reflected in and reinforced by information systems and the ethical implications regarding the 'control' of agents via their interactions with them. Digital spreadsheets can be generally viewed as one of many knowledge artifacts or what Giddens calls 'containers' (reports, meetings, presentations, networks, dashboards, websites, and so on) where information is stored, organized, transformed, and transmitted by systems [10]. How digital spreadsheets are appropriated in

specific ways has received little to no attention in the literature. The present article attempts to fill this gap. It looks at how the properties of spreadsheets are employed and created to influence workers and invoke intended outcomes that ultimately reproduce and produce power relations within modern organizations.

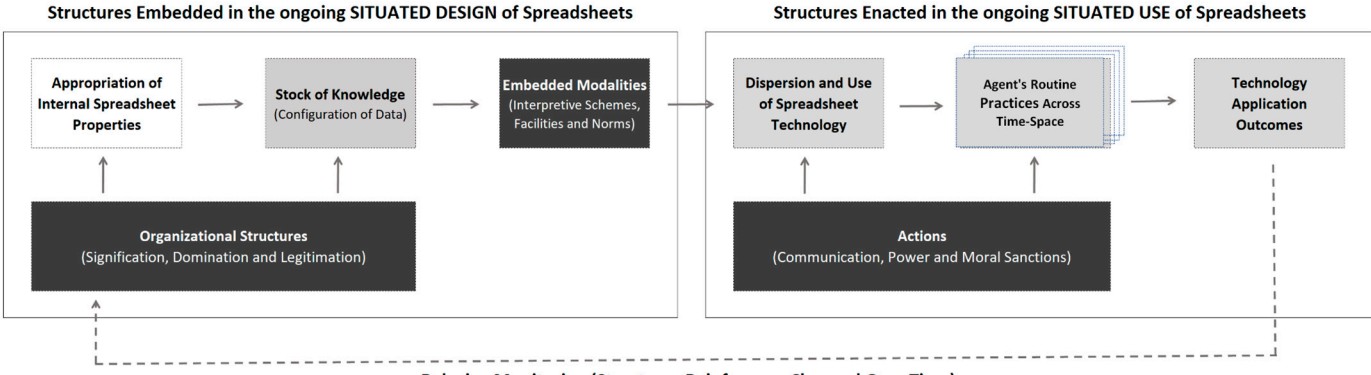

**Figure 3.** Digital spreadsheets-in-practice conceptual model. Adapted from Orlikowski and Giddens [1,3].

## 2. Methodology

The focus of this article is on revealing how the capabilities and features of spreadsheet applications, the 'properties' of spreadsheets, make them reconfigurable tools for representing structures in different contexts. In the analytic process, spreadsheet design practices are identified that constitute and technically work as Giddens' three structural modalities of power, as discussed earlier. These roughly translate as: (1) construction of interpretative schemes for meaning making and rationalizing courses of action, (2) exercise of authoritative power for automated decision making and application of resources, and (3) enactment of moral sanctions and social norms to discipline people's work. The three modalities are viewed as overlapping and reinforcing layers or dimensions of structure supported by a 'stock of knowledge' that is constructed using the fundamental gridmatic architecture of spreadsheets and certain common spreadsheet properties and design practices.

Grounded in the deconstruction of digital spreadsheet artifacts, the framework presented in this article reveals how spreadsheet properties imbue recorded information with significance, purpose, and action. Specifically, our structural framework posits that the three modalities draw on three sets of spreadsheet properties: (1) formulas, (2) analytical models, and (3) logical functions (Figure 4). It demonstrates how organizations use spreadsheets to systematically transform information, produce knowledge, predetermine decisions, allocate resources, and apply moral sanctions for the purpose of enabling and constraining conduct.

This article is less concerned with revealing the practical uses and effects of spreadsheet technology and more concerned with elaborating on what organizations do with spreadsheets to produce and reproduce organizational structures [3,11]. The focus is on characterizing how digital spreadsheets have been refined over time to systematically embed meaning in organizational structures. This is demonstrated through vignettes of situated 'technology-in-practice' that demonstrate how the managerial work of an organization enacts the architecture, formatting, data validation, formulas, macros, analytical models and other properties that constitute digital spreadsheets.

The three structural modalities are regarded as a framework composed of the sequential and reciprocal process steps or layers of analysis that have the effect of constructing new and reinforcing current relations of power within modern organizations. The key outcome presented in the conclusion of the present article is a proposed conceptual framework that encompasses these dimensions and provides the foundation for future practical research in information systems. The framework consists of a typology for characterizing the various

properties of digital spreadsheets according to how they are used in executing each of the structural modalities. In the following sections of the present article, each of the four main components and their respective spreadsheet properties are demonstrated

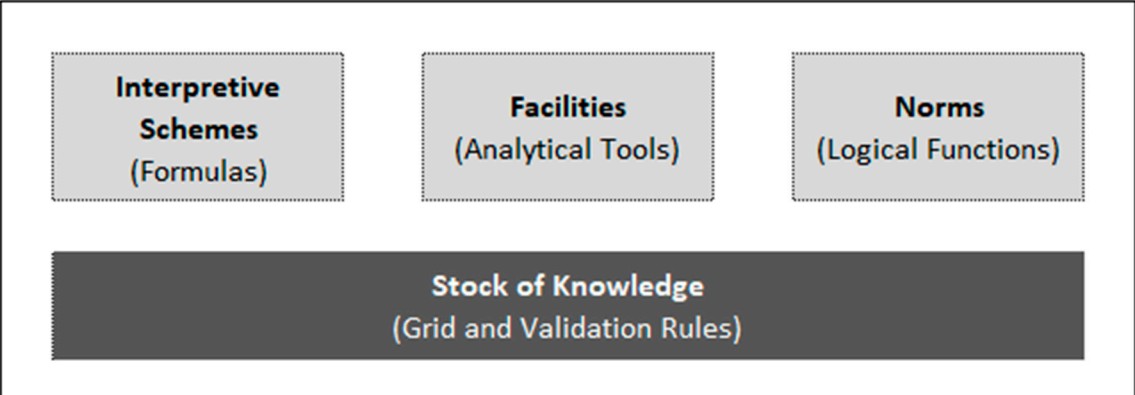

**Figure 4.** Three structural modalities and corresponding spreadsheet properties. Adapted from Giddens [1].

### 3. Configuration of Information into a Stock of Knowledge

Spreadsheets work like records that organize knowledge and meaning. They function as systems for the collection of data into databases that purposefully construct the organization's reality and legitimacy. They developed their basic gridmatic structure from paper-based book-keeping ledgers that recorded and tallied commercial transactions as debits and credits [12]. Columns representing types of expenditures were listed at the top of the ledger and transactions were recorded in the left margin. Therefore, spreadsheets combined the basic categorization power of lists with the cross-referencing capabilities of tables to create cells of meaningful information. This provides familiar ways to convey relatively simple or complex information as well as the visual structure of the data so that new dimensional relationships are produced.

However, spreadsheets are not perfect, unequivocal storehouses of neutral and error-free information. Spreadsheets embody the interests of the designers who create and maintain them and legitimate their value. They are continuously produced to influence our perception of truth and reality by shaping how information is represented. Spreadsheets do not only record reality, they make it. The designers of spreadsheets control the information that is central to the collective memory and culture of an organization. Spreadsheets, as structural modalities, aim to describe, communicate and legitimate certain representations of reality. Spreadsheets viewed as archives are not passive collections of data, but as Schwartz and Cook explain, a spreadsheet is "a reflection of the needs and desires of its creator, the purpose(s) for its creation, the audience(s) viewing the record, the broader legal, technical, organizational, social, and cultural-intellectual contexts in which the creator and audience operated and in which the document is made meaningful" [13].

The fundamental power of spreadsheets rests in their gridmatic framework which combines the categorization power of lists with the cross-referencing capabilities of tables to create cells of meaningful information [14]. Spreadsheet applications contain properties (features and capabilities) that have developed over time to facilitate active management practices. These include controlling what information is included in their document construction. Data are carefully assessed and intentionally selected to construct and maintain the record. This is not conducted objectively or neutrally; the techniques for organizing data as it is manually entered or imported into a spreadsheet determines its structure, what it becomes and how it is used. Therefore, spreadsheets constitute an understanding of reality by organizing and privileging some information and minimizing or even erasing other information.

### 3.1. AT&T Illustration

The common practice of Valuation Modeling with spreadsheets, for example, is simply an extension of spreadsheets' innate ability to produce power through storing information about the tangible assets and debts of organizations. To give a real-life example, consider the well-known case of when AT&T's Ma Bell's monopoly on telephone service in the United State and Canada was relinquished due to an antitrust lawsuit. In 1984, the company was split into seven new regional operating companies commonly referred to as the 'Baby Bells'. The 3.2 million holders of stock in the original AT&T were given stocks in the spun off companies. The basis of these shares was the economic value of each Baby Bell, which was essentially calculated as the difference between their total assets and liabilities, the balance sheets of the companies.

Federal Communications Commission (FCC) records from the time discuss the key principle that stock values for the Baby Bells should be equal to their 'economic value' or 'book value' of all assets transferred to the new entrants [15]. The companies' book values were thus the "practical surrogate" of an enormous inventory ordered by the Federal District Court in Washington called the Bell System Asset Assignment Detail Work Plan [15]. Spreadsheets that contained inventories of AT&T's assets were developed that included information from existing records as well as physically counting certain items when records were not available. Using the list, assets were then divided among the divested companies "based on the principle of sole or predominant use" [15]. The numbers of "transmission, switching and plant facilities—including cables, poles, buildings, motor vehicles, office equipment and furniture [were] assigned to an operating company or to AT&T according to which entity uses them most" [15].

The AT&T breakup example highlights the concept that the fundamental power of spreadsheets rests in their data gathering, storage and organization abilities and the knowledge they produce about the information they contain which are then represented as indisputable facts they communicate about reality. This subtle and seemingly passive ability of the act of how recording information conducts power is achieved through the imbued objectivity and authority ascribed to the knowledge produced by spreadsheets. They are imbued with authority claims (truth, proof, importance, endorsement, validity, directive, etc.) by the structures of the organizations that create and use them and by outside agents such as government regulators, special interest groups, customers, industry competitors and suppliers who reinforce and validate them through their widespread use. As Schwartz and Cook assert, the fundamental power of spreadsheets can be found in how they function as records and "are about imposing control and order on transactions, events, people, and societies through the legal, symbolic, structural, and operational power of recorded communication" [13].

### 3.2. Controlling Data Entry with Validation Rules

The built-in properties for controlling and restricting how users enter data into a spreadsheet are called 'validation rules' in Microsoft Excel (IBM, New York, NY, USA). How they are drawn upon varies, but generally these rules are meant to prevent data entry mistakes and maintain consistency so that formulas within cells can perform calculations on error-free data. In Microsoft Excel (IBM, New York, NY, USA), they can take the form of data 'formats' that are activated as a user types information, such as controlling the layout of a date or email address. Microsoft Excel (IBM, New York, NY, USA) allows designers to write custom formulas to be used in these formats based on data entered in other cells and to customize input and error messages. Validation rules can also take the form of dropdown lists that provide a limited number of possible choices which can be static or dynamically linked to data entered in other cells and lookup tables. For example, a spreadsheet might generate a list of customer names in a dropdown list by looking up information in a different table. Sometimes the actual value in the cell is entered differently compared to the options displayed in a dropdown list. For example, a worker's name in a dropdown list might be translated into the corresponding worker's identification number

in the cell. Multiple dynamic custom data validation rules can even be nested in one cell to verify data input for a wide range of complex possibilities and cells can be locked so that most users cannot change or delete the rules.

Data validation is a feature found in most spreadsheets, databases and information systems, especially when they are created for business use and maintained among a group of workers. For obvious reasons, the technique is a general best practice of effective spreadsheet design. However, the application of validation rules is not neutral even though they often go unnoticed by users. As the name suggests, they are 'rules' and they display the signs of structures (values, standards, requirements, guidelines or policies) that through interactions with a spreadsheet are reproduced and enable the system of the spreadsheet to operate as a modality or medium of power. Validation rules also function to privilege information by setting aside presumed non-relevant data and focusing on data that are sanctioned by the organization, which is essential for establishing the supposed validity of the data and for enabling subsequent layers of more complex data analysis and interpretation.

### 3.3. Employee Database Illustration

When Excel is used to create a simple employee database, for example, each record (row) in the table will represent a different employee listed by name and the columns are fields that represent the attributes or characteristics of the individuals for which data are recorded. The information assembled by the spreadsheet allows the organization to look up and compare employees, aggregate the information and produce reports. It is important that data are entered consistently and without errors, or the reports will not produce reliable insights. As such, validation rules are commonly used throughout such spreadsheets to administer the basic format of data inputted into cells such as names, email addresses, telephone numbers, dates and so on of each respective employee. Many cells such as gender, marital status, and nationality restrict data entry to predefined dropdown lists. Larger dropdown lists that change frequently are derived from lookup tables located on other tabs in the workbook, such as job title, supervisor and division. On its surface, the collection of different validation rules included in a spreadsheet compose a purposeful scheme for preventing and rejecting incorrect data from being recorded.

The spreadsheet's validation scheme produces 'clean data' which allows for automatic filtering to search through and group records by division, supervisor, date hired, years of service, salary, and any other fields germane to understanding the information. As the different filtering options are activated, the 'population' of employees represented in the dataset or stock of knowledge is effectively segmented for analysis. Thus, the automated validation programmed into the spreadsheet serves as the essential first step of applying interpretive schemes. As filters are applied, certain cells containing Excel's built-in functions for databases dynamically calculate statistics used to produce a straightforward interpretation of the information such as head counts, salary distributions, and payroll breakdowns.

Giddens emphasizes that no matter how apparently quantitative and empirical an analytical model may appear, underlying them are always assumptions, beliefs and prior interpretations (i.e., structures) grounded in their unique social context which must be included in the examination of their use [16]. In the simple employee database example, data validation techniques are applied to convert the data into a recognizable form necessary for the intended use of the spreadsheet—to produce understandable results from elementary statistical analysis. The 'correct' way to enter data is grounded in the situated use of the spreadsheet. Why only certain job titles are made available in the drop-down list, how they are written, and their meaning is rooted within the structures of the organization and represent "active sites where social power is negotiated, contested, confirmed" [17]. The same spreadsheet transferred to a different company would be changed to reflect different values and rules regarding what information about employees is considered most important to include in the database. An insight reiterated by Ketlaar:

Archives echo what their creator has meant and what the user of the record wants the document to tell him or her: expressions of societal power. Every interaction with the record by creator, user, and recordkeeper is enforced by power. Each of these activations leaves fingerprints which are attributed to the record's meaning. [18]

*3.4. Meaning*

The structures reflected in the spreadsheet are imposed on the user through the numeric values and programmed formulas embedded in them. In many applications of spreadsheet databases, lookup tables are locked and/or hidden from users to prevent the definitions or the formulas from being changed by anyone other than the designer. In most cases within organizations where spreadsheets are commonly shared within or across teams, a spreadsheet's validation and meaning-making schemes cannot be modified or even made visible without permission from the designer. The end effect is that the stock of knowledge produced by the spreadsheet is controlled by the automated configuration of the interpretive scheme that renders the data meaningful. This whole process is initiated through the seemingly trivial interaction between the user and a spreadsheet's dropdown lists, a validation rules technique that has been a basic feature of Microsoft Excel (IBM, New York, NY, USA) for decades.

Spreadsheet validation rules can be utilized to 'privilege' how information is entered and organized in a spreadsheet which has the effect of reducing the data and thus narrowing the range of possibilities of a user's interpretations and their subsequent actions. Validation rules in the form of dropdown lists represent the restricting and converting of inputed information into numerical values to enable the spreadsheet's analytical structure. The result is that spreadsheets work to code, translate, orient and give meaning to information to construct stocks of knowledge that reproduce and produce a purposeful narrative. Dropdown lists actively formulate the information according to the organizational structures reflected in the spreadsheet, and their meaning and power is not clear unless considered within the context of their use, their situated state or position. To use Michel Foucault's terms, the underlying principles or 'order' of a spreadsheet represents what can be said, when, and with what authority, a "system of enunciability" that defines the expression of statements or the "statement-thing" [19] (p. 2). A spreadsheet determines the conditions of the possibility of its construction, emphasizing less the static collection of data but rather a stock of knowledge that sets the relations and meaning produced by it.

## 4. Configuring Interpretive Schemes with Formulas

The previous section examined how the organizing gridmatic framework of spreadsheets can be used to produce an initial layer of meaning by reducing data as it is entered to privilege what information is recorded and how it is maintained as a stock of knowledge. The point was made that spreadsheets combine an easily viewable display format in a simple geometric space. A spreadsheet's base capacity to organize data in a spatial tabular format (rows, columns and intersecting cells) enables an analysis of spreadsheet data. This format essentially turns the data in cells on a sheet into specific objects that are treated as equal entities locked in place in a series, which creates an internal connection or relationship between each individual record to each other and to the dataset as a whole. The discrete location of the data in each cell in a sheet has its own unique coordinates that can be easily referenced. This structure allows a spreadsheet to sort quickly and then dynamically associate, compare, and compute the data within the cells that constitute segments of a dataset by using formulas to interpret it and produce meaning.

Empowered by the fundamental tabular structure of spreadsheets, formulas are able to reference any other cell or range of cells by their X and Y coordinates shown as a combination of column letters. These coordinates give cells a 'symbolic name' to stand in for the data as variables in formulas in combination with Excel's pre-programmed functions rather than the actual values of cells themselves. For example, the top left cell in

every spreadsheet is always named 'A1' and the first three cells in the first row would be referenced as 'A1:A3' in a formula. If we want to add the values in these cells, we would write a formula by selecting an empty cell and then entering the proper syntax for the basic SUM function in Excel's formula bar: "=SUM(A1:A3)." Excel also allows designers to bind any cell range as a custom name of their choosing, thus creating common objects to use as arrays of data that can be passed to formulas.

This organization of the information into cells in a spreadsheet serves as the foundation for overlaying a sequence of interdependent levels of analysis that define more complex relations and meanings. Giddens calls the first such layer examined in this section an 'interpretive scheme'. Interpretive schemes are defined by Giddens as semantic rules that "involve the encoding of information in symbolism . . . which are transformative in character" [1]. Interpretive schemes can also be viewed as collective mental models used to make sense of the data and communicate how people should respond in particular situations [1]. Giddens' premise here is that the decisions workers make, and their social interactions, depend on how the information is formatted, framed, characterized and evaluated—the 'significance' of the information within its situated context which can have a regulating impact on decision making and what people do, therefore serving as the medium of power.

The main way interpretive schemes are embedded into spreadsheets are as programmable formulas, as was shown earlier when discussing the illustration of the simple employee database. Formulas have the capacity to calculate values and generate new information based on the relationships between values in other cells. Formulas represent the process, actions or methods needed to evaluate and run statistical tests on a dataset. Because spreadsheet cells are uniquely named objects and not the data they hold, formulas can be easily copied, moved, tweaked, replaced, and so on to produce different results regardless of the dataset. This gives interpretive schemes in Excel their own visible spatial structure that is independent of the data and allows formulas to be electronically distributed as pre-built meaning-making schemes within and between organizations across time and space.

### 4.1. Loan Valuation Illustration

Building on the practice of valuation modeling with spreadsheets discussed previously, an elementary example of how formulas create meaning is the way financial institutions use spreadsheets and other information systems to appraise the explicit value of auto loan applications submitted by their customers. In standard spreadsheets used in the banking industry for years to make auto loan approval decisions, cells in the matrix contain formulas that reference data stored in the spreadsheet to calculate numeric values for the abstract criteria used to scrutinize the merits or risk of a borrower's loan application. The criteria could include fields such as the applicant's self-reported income, current debts, monthly expenses, real estate assets, number of dependents, requested loan amount, term, down payment and so on. Other information from outside sources is also entered by the loan officer, such as the applicant's credit score and the estimated value of the vehicle after purchase. Formulas are then used to summarize these variables to figure out the maximum loan amount that is approved with certain conditions.

The formulas encompass and substitute the work of appraising loan applications. For modern banks, loan applications are now submitted online and produce a decision in minutes without the involvement of loan officers. They also produce what Deringer describes as the "impression of objectivity, a certain accuracy, because they look complicated, . . . create these elaborate structures, and produce what seem to be incredibly precise answers" [20]. However, the example of the auto loan appraisal tool is not neutral. The encoded formulas of the spreadsheet are underpinned by the intangible espoused structures of the bank that created and uses it: what data can be altered or not (validation), which variables are privileged to begin with, how the deemed relevant variables are explicitly calculated, and the amount of weight each variable is given in determining the maximum

loan amount are all preset criteria determined by the banks guidelines, policies and rules for authorizing this type of loan.

The way in which the formulas are appropriated to transform and analyze data entered into the spreadsheet are purposeful and mediate the 'right' procedure of appraising loans that align with organizational structures. In essence, formulas allow the spreadsheet to represent the organization in guiding the cognitive thinking of workers when interpreting data to limit how the information can be understood and to automatically validate decision making in a certain way. The spreadsheet's abstract interpretive scheme embodies a common strategy for appraising loans in which loan officers interact with the situated 'thing' that is the spreadsheet as a matter of routine procedure, and this interaction disciplines what they do in their everyday practices—what happens in terms of a loan application being declined or approved with particular terms every day at the bank. The spreadsheet is a situated artifact that symbolizes the communally agreed-upon way of carrying out the work of loan application appraisal [21]. When the spreadsheet is widely disseminated throughout the bank, it has the additional effect of reinforcing consistency in the patterns of enacted conduct among the loan officers as a group; it structures their routine practices and creates social coherence.

*4.2. Framing of Information*

Spreadsheet interpretive schemes such as the above loan authorization tool and the earlier-discussed related valuation techniques evolve over time and become accepted and legitimated through their use. Spreadsheet artifacts are "epistemic objects, which are objects that gain situated meanings within the process of being used in knowledge work" [17] (p. 40). They are commonly shared, updated and validated through communities of practice, reported on in publications and taught in schools. Thus, their influence extends far beyond a single organization, and they reflect the capability of formulas in Excel to call upon information stored in cells by their constant symbolic names, which empowers the dynamic and temporal qualities of interpretive schemes. As the source data change, formulas can immediately recreate, repeat and re-present the results of preset calculations and thus the effects of interpretive schemes. The results of a spreadsheet are temporary, representing only the point in time at which the spreadsheet was last refreshed or last updated. Furthermore, the cell references in formulas are typically relative—they are automatically updated to reflect the shifting locations of the formulas in a spreadsheet. The cells referenced in a formula may themselves be expressions of other formulas. Thus, formulas can be constructed from multiple prerequisite sub-formulas in order to allow designers to create complex analytical models in which a change to one deeply nested formula that represents a constant can have a cascading effect on the returned calculations of most other formulas in a spreadsheet.

In their ability to structure information according to preset interpretive schemes, spreadsheets are able to constitute understandings of organizational reality in a way that includes or privileges some interpreted meanings and foreclosed on others. Where validation rules operate to privilege data, interpretive schemes similarly 'control' the automatically generated analysis of stored information through presupposed analytical techniques rendered as embed formulas. A digital spreadsheet performs the organizing and categorizing of a dataset in accordance with a specific set of rules and standards that constrain the way stored information is organized, transformed, interpreted and acted on. These value-laden rules determine how a spreadsheet's logic is structured and the conditions through which data interpretation is framed. They are often subtly used to govern interactions with spreadsheets and are largely a reflection of organizational interests and culture.

## 5. Making Decisions with Analytical Tools

In the previous section, it was shown how spreadsheets are commonly purposefully designed with certain locked-in interpretive schemes that produce knowledge, which is

often diffused throughout an organization. Spreadsheets construct sites for the dispersion of 'truth' based on a set of embedded structures largely composed of 'rules' governing data interpretation that reflect organizational structures in both the immediate social context (purposeful practices) and the intentions of a broader field of power (managerial interests). This section explores how the previously discussed layer of interpretive schemes fixed in a spreadsheet underlies the intertwined structural modality that Giddens calls 'facility'.

It is Giddens' premise that information systems construct knowledge, produce meaning and shape reality in a particular way to induce users into making what organizations deem as 'correct' decisions [1]. The information communicated by spreadsheets underscores the rationalization, justification and legitimation of courses of action based on a set of rules. In short, knowledgeable agents draw on interpretive schemes to bring their decisions into alignment with organizational structures.

Giddens also argues that knowledgeable agents' understanding of what they should do runs side-by-side with their capacity to allocate resources. Interpretive schemes give the "rights to the individual to demand resources from others" [1] and their "capability to influence events (make decisions) depends on resources they can mobilize" [1] (p. 33). Spreadsheets construct information about the abstract 'things' they list, count, measure and track, while at the same time serving as 'facilities' for how to call on these things as resources to empower actions. As Giddens argued, material resources "become resources only when incorporated within processes of structuration" through spreadsheets and other information systems [1] (p. 33).

### 5.1. Excel Plug-Ins

In the earlier AT&T illustration, it was shown that the interpretive scheme used to record and calculate the assets of AT&T was then used to distribute the same resources to the newly created companies. Similarly, a loan appraisal spreadsheet not only allows loan officers to interpret data and make a decision, as discussed earlier, it empowers them to transfer the bank's resources in the form of money to the recipients that are selected during the process. The stock of knowledge and interpretive schemes embedded into a spreadsheet are in essence the figurative building blocks for more complex analytical models that attempt to influence and automate decision making in organizations and reinforce wider social practices that can become widely prevalent across an entire industry.

According to Will Deringer, former investment banking analyst at the Blackstone Group, spreadsheets are "considered by some to be the most important invention in the history of modern finance" [1] (p. 38). Digital spreadsheets were particularly central to the "financial revolution" of the 1980s. They became known for their formulas that could perform incredibly complex 'what-if' scenarios that were beyond what programmable calculators could do. Campbell-Kelly likens "digital spreadsheets to a computer game for executives . . . they simulate real-world situations, and you can change the parameters to see how different financial scenarios play out" [1] (p. 38).

In addition to hundreds of specialized financial functions added to Excel over the years, Microsoft Excel (IBM, New York, NY, USA) has been equipped with a wide array of pre-programmed 'plug-ins' originating from the field of decision analysis, which included what-if analysis (data table, scenario manager and goal seek), forecast sheet, statistical analysis tools and Solver (Linear Programming). Starting back in the 1980s, corporate raiders, for example, began to use these tools to quickly test various scenarios where one company could take over another company. In a time of rapidly changing interest rates, spreadsheets could also quickly recalculate interest and principal payments on various loans and determine monthly payment schedules. This made whole new forms of financial transactions possible, such as Commercial Mortgage-Backed Securities (CMBS), Credit Default Swaps (CDS), and Collateralized Loan Obligations (CLO). These new 'schemes' were at the center of the 2007 subprime mortgage banking crisis, as reported in the book *The Big Short: Inside the Doomsday Machine* by Michael Lewis [22].

*5.2. WorldCom Illustration*

To substantiate the above claims and illustrate the power of widely used spreadsheets in society, consider David Faber's account of the notorious WorldCom case [23]. In the mid-1990s, as the Internet and its World Wide Web were gaining the public's attention, Tom Stluka created scenarios for the Internet's growth on an Excel spreadsheet. Stluka was an engineer for UUNET, a popular Internet service provider (ISP) that was taken over by WorldCom in 1996, a long-distance voice telephony company. He regularly developed estimates for data traffic based on a spreadsheet model he had created. Stluka's CEO, Kevin Boyne, would often encourage him to increase his traffic forecasts. Boyne wanted his suppliers of fiber optics and other new telecom equipment to increase their production so that supplies of the glass conduits and routers would be sufficient and prices would be driven even lower due to an abundance of supply. The so-called 'big lie' emerged that the Internet was doubling in size every 100 days. By citing Tom Stluka's internally developed Microsoft Excel spreadsheet (IBM, New York, NY, USA), Boyne was able to circulate the meme. However, in working with the analytical model, Stluka was pressured by Boyne to simply assign variables with various parameters to produce "whatever we think is appropriate".

It became known as the 'doubling meme' and spread quickly through the media that was rapidly expanding its interest in technology. Traditional companies such as AT&T and TCI invested heavily, and new entrants such as Enron, Global Crossing, and Tyco entered the telecommunications market. Stocks of networking companies such as Bay Networks and Cisco took off. As the dot.com boom took off, the doubling meme became the doubling mantra. This case of the spreadsheet that changed the telecommunications environment of the 1990s operated initially within the WorldCom operation. Then, it produced results that diffused throughout the Internet industry and investment community. The story became a bit of an urban myth, but it points to its rhetorical value as it circulated through the technologically driven economy of the 1990s 'Bull Run' era.

The WorldCom case illustrates how power emerges from work-related interaction with digital spreadsheets that are derived from the context of the organization and industry within which they are produced—the ways in which they are enacted [24]. Stulka's spreadsheet model induced a recursive flow of knowledge and action that produced a total industry transformation and ultimately resulted in enormous negative economic consequences caused by the catastrophic breakdown of the U.S. economy. The structuration of the information through the spreadsheet works as a modality for producing and reproducing meaning and imbued truths in the organization and throughout the industry, which ultimately justified misguided investment strategies on a large scale.

The WorldCom example shows how Microsoft Excel's (IBM, New York, NY, USA) built-in decision analysis tools can extend the 'results' of interpretive schemes to stretch control over the allocation of resources over space and time by managers who are often not present when and where a spreadsheet is being used. Furthermore, spreadsheets may function like automated and prescriptive decision-making systems (DSS) that provide predetermined solutions to routinely experienced problems. Excel plug-ins such as Power Query can completely automate real-time data updates, cleanup, reduction, reorganization and merging. Macros will automatically run when a spreadsheet is opened or refreshed to automate spreadsheet interactions, which users would otherwise need to do themselves. In a typical spreadsheet DSS, users are given a limited number of input choices used to configure the analytical model and then after selections are made, the results are outputted as a report without the data or the calculations in the spreadsheet ever being revealed to them during the process.

*5.3. Provisioning Things*

Spreadsheets facilitate the collection, categorization, and systematization of flows of data that aid the efficient management of resources to pursue organizational objectives by remotely arranging and integrating employees' work activities. The spreadsheet provides

the means to distribute resources and provides a key nexus of power in organizations when only certain individuals are empowered to use or apportion resources. O'Regan's quote below alludes to this power of a spreadsheet to organize quickly and give legitimation to operational decisions:

> As much as oil and water, our lives are governed by Excel. As you read these lines somewhere in the world, your name is being dragged from cell C25 to D14 on a roster. Such a simple action, yet now you'll be asked to work on your day off. It is useless to protest. The spreadsheet has been printed—the word made mesh. [25]

A spreadsheet is a proxy for the unequivocal oversight and direction of managers when they cannot be present. Think of a military structure where the chain of command signifies the power to assign duties to subordinates or allocate provisions such as food, water, and ammunition to different units. The development of different ways to track and inventory resources and how to provision the same resources are inseparable. Spreadsheets can also organize the time-space sequencing of events and actions when organized as Gantt chants, time-tables and other time management techniques [26]. A stock of knowledge constructed in a spreadsheet facilitates the utilization of an interpretive scheme that in turn validates decisions regarding the allocation of the very same things tracked by the database. Interpretive schemes (knowledge) and the allocation of resources (power) are interwoven together in a recursive loop shaping the boundaries of the autonomy of agents to counter hierarchical decision-making processes.

## 6. Invoking Social Norms with Logical Functions

In the previous section, it was discussed how interpretive schemes embedded in spreadsheets provide the narrative, justification or rationalization for the use of facilities by organizations, which involves the control and allocation of resources including the management of the activities of workers. However, these practices also involve the notion of judgement and accountability where the same information used to track and authorize the use of the object things recorded and analyzed in a spreadsheet are also used to approve and disapprove of agents' decisions concerning those things. As Roberts and Scapens emphasize, the values of expected behavior in accounting include "the rights of some people to hold others to account for their actions ... communicating notions of what should happen, and it is only on the basis of these notions that sense is made of what has happened" [20].

The earlier loan appraisal system example also shows how a spreadsheet's interpretive scheme can be set up to judge the worth or value of a bank customer. This procedure is similar to the common business practice properly known as Customer Lifetime Value (CLV) in which spreadsheets are used to estimate the value of different segments of customers based on how much they are expected to contribute over a period of time and assign them a score that symbolizes their assessed desirability. These spreadsheets are broadly purported to be useful for identifying customers that need the most improvement or are the most profitable and implement strategies to get the 'low-performing' customers to change their purchasing behavior in some way. This is achieved by developing a matrix of weighted input criteria or assumptions based on quantifiable factors that theoretically represent the characteristics and behaviors of customers, including acquisition cost, average order value, purchase frequency, gross margin, retention rate, churn rate, average lifetime period and more.

The authors of the present article have witnessed similar banking approaches appropriated by most colleges which use secretive algorithmic formulas to score, rank and select prospective students. The formulas typically draw on data submitted by students via online application forms such as college placement test scores, teacher recommendations, class rank, level of interest, extracurricular activities, and other sociodemographic characteristics. Analytical models then formulate the data into virtual portfolios for each applicant which are used to compare candidates to each other and past students to appraise their relative

'value' or 'worth'. In other words, these enrollment systems operate to identify the most 'favored' candidates based on the likelihood of them enrolling and successfully graduating, and thus paying the full amount of projected tuition. This information can then be used to help inform the decision-making process along with many other assessments to determine which applicants 'make the cut' and are accepted into a college.

The recorded performance of the enrolled students over time then becomes an ongoing source of information to help evaluate and optimize the decision-making systems. Once again, as new prospective students apply, they are classified according to analysis of surveillance data on current and past students. The applicants are reflexively aware of the criteria the systems use and attempt to 'game the system' in ways to produce statistics that give them the appearance of favorable candidates. The CLV spreadsheets compel applicants to meet the expectations set by school administrators and thus have disciplinary effects that shape, align and normalize those students with the structures the spreadsheets embody.

*6.1. Logical Functions*

The above CLV examples illustrate how in spreadsheets, a third symbolic layer of interpretation and meaning based on social norms is overlaid onto the data by using some simple techniques involving logical and conditional functions built-in to Microsoft Excel (IBM, New York, NY, USA). Using logical functions, a spreadsheet qualifies a dataset in accordance with the assessment criteria in addition to its actual original recorded information. This results in the automated production of knowledge about the worth or performance of records in the dataset in relation to each other or an expectation. Essentially, a spreadsheet uses simple logic embedded by its designer to judge the dataset and by extension the performance of the workers the data represents.

The aforementioned capability of spreadsheets to make judgments is enabled by built-in conditional functions in digital spreadsheets, which are used to query datasets based on simple Boolean logic. As in many programming languages, the IF function is used in Microsoft Excel (IBM, New York, NY, USA) to evaluate whether a condition is true or false. It uses an operator such as "=", "<" or ">" to compare the fit of values in a data set to a given criteria and perform actions as a result. Essentially, it asks: "Does the referenced value satisfy the provided condition?" Next, it runs the corresponding action: "If true then run script X, otherwise do nothing or optionally run script Y." By adding conjunctions such as AND, OR, XOR, and NOT to the IF function in Microsoft Excel (IBM, New York, NY, USA), multiple combinations of conditions can be tested. Microsoft Excel (IBM, New York, NY, USA) has also combined IF with other basic mathematical calculations to create new functions such as SUMIF and COUNTIF that only allow certain calculations to be performed when the stipulated criteria are satisfied.

The related VLOOKUP, INDEX, and MATCH functions in Microsoft Excel (IBM, New York, NY, USA) also operate on true/false logic to compare values and return a result. VLOOKUP (Vertical Lookup) allows the spreadsheet to 'lookup' and retrieve data based on information from the dataset. Taking advantage of its tabular format, a spreadsheet can use VLOOKUP to examine a column of sorted data for the first value that exactly or approximately matches the lookup criteria. It can then return a separate value from a neighboring column. Alternatively, HLOOKUP (Horizontal Lookup) allows the spreadsheet to search through rows of data. Additionally, by combining the INDEX and MATCH functions, a Microsoft Excel (IBM, New York, NY, USA) spreadsheet can look up values both horizontally and vertically within a data set and return information about the values.

One common use of the previously described conditional functions is to classify data and assign categories. For example, an series of nested IF functions in a formula can transform a percentage test score into a letter grade, as in: "=IF(Grade>=90,"A", IF(Grade>=80,"B", IF(Grade>=70,"C",IF(Grade>=60,"D","F")))))". Alternatively, a spreadsheet could use VLOOKUP and a grade key table to accomplish the same task: "=VLOOKUP (Grade,GradeScale,2,TRUE)". If a teacher wants to grade on a curve, the mean of the total

scores is found using the AVERAGE function, and then this relative output is referenced if the IF statement: "=IF(Grade>=Average+20,"A",IF(Grade>= Average+10,"B",IF(Grade>= Average,"C", IF(Grade>=Average-10,"D","F"))))".

Operating on the same Boolean logic, Microsoft Excel (IBM, New York, NY, USA) also has built-in Conditional Formatting rules that, once set up, allow a spreadsheet to automatically reformat a cell's properties to correspond to its assessed value. As in the earlier example, conditional formatting allows a spreadsheet to instantaneously change the background color of a cell containing a score of 90–100 to blue, 80–89 to green, 70–79 to yellow, and so forth, depending on the rules without the use of a formula or programming. Similarly, true/false logic is used in Excel to validate or constrain the data that can be entered into a cell through dropdown lists, checkboxes, and other form validation tools.

### 6.2. Call Center Illustration

One of the authors of the present article has nine years of experience managing and working in call centers. In these organizations, intense agent monitoring is believed to result in high performance. Thus, the work of call center operators is highly meditated by Computer-assisted Telephone Interviewing (CATI) information systems. The moment an operator starts his or her shift by logging in with their unique credentials, the CATI systems automatically generate a queue of customers to dial. Information about who they are calling and a script to read is then shown on an operator's screen as they listen to the phone ringing in their headset. Every interaction the operator has with the information system is recorded before, during and after the calls.

On average, call center operators will make thirty or more unanswered calls before talking to an actual customer. The CATI system tracks the time and duration of calls, and most importantly, the outcomes. A dropdown list will usually provide operators with preset categories for coding calls by their resolution (call codes): no answer, disconnected, hang-up, call back later, etc. Through this interaction, the system creates a historic record of the outcomes of every call. In the less-often occurrence when an intended customer actually answers a call, a different set of outcomes is used depending on the type of business.

In a debt collection center, for example, the top goal is for operators to convince delinquent customers at risk of defaulting on their loans to immediately pay their past due bills over the phone with electronic bank account transfers. Another desirable call resolution is for a customer to sign up for automatic monthly payment plans. A less optimal outcome is when a customer promises to pay later. An undesired negative outcome of a call is if a customer outright refuses to make a plan to catch up on their past-due payments. Operators are continuously trained and coached in many tactics to persuade delinquent borrowers to make up their missed payments in order to increase their collection rates.

The call codes represent the call center's system of accountability. There is an entire structure of enunciated values underlying this seemingly inert list that provide an 'electronic eye' into each worker's performance based on the imparted value of each call code [27]. The individual results of the call codes along with other key performance indicators (KPIs) involving time are linked to spreadsheet models that allow managers to assess the relative productivity of operators against established expectations. In debt collection center, spreadsheets are used to calculate many KPIs: Revenue per Successful Call (RSC), Average Handle Time (AHT), First Call Resolution (FCR), and others that are then combined to compose an overall Collector Effective Index (CEI). Formulas in the spreadsheet apply Boolean logic to mark which operators are above or below an ideal set of standards (benchmarks) for the KPIs, and also automate the reordering and formatting of the lists by the relative rank of each operator, placing the judged 'best' performers and the top of the list and 'worst' at the bottom of the list. The aggregate of the productivity of all operators composing each team are 'rolled up' to produce an overall measure of the teams' relative performance. These reports are then used by higher-level supervisors to evaluate, rank and predict the performance of teams and team managers.

The spreadsheet analysis of call center data not only measures performance, it also elicits the performance, or at least a semblance of compliance with imputed norms associated with meeting expectations [28]. Recording and analyzing the data is closely linked with anonymously 'watching' what call center operators are doing through statistics and then using this information not only to 'judge' their performance, but also correct and praise their conduct to influence them into adopting behaviors that are aligned with organizational structures [29] (p. 176). On inspection, operators must show their managers to be continuously improving themselves to become more productive as measured by the call codes and the time stamps recorded by the CATI system and interpreted by spreadsheets.

The call center example shows the mutual connections between the collective production of information and institutional analysis with norms and interpersonal forms of surveillance [30]. The definitions of the call code criteria represent certain irrefutable and unquestionable measures of performance based on the structures (values, principles and standards) privileged by the organization. This objectification of operators as statistics can then be used to justify managerial strategies for encouraging positive and curbing undesirable behaviors, disciplining workers through reward and punishment schemes. High-performing operators receive public praise, are awarded incentive pay, are scheduled for more hours and are given the choice of which shifts and days they want to work. On the other hand, low-performing employees are subject to embarrassing coaching sessions or retraining, have their hours reduced, or may not be called back to work at all.

It is a common tactic in call centers for team managers to regularly share spreadsheet performance reports with operators and even post them publicly for everyone to see. This is believed to induce competition among operators and teams. Operators are expected to contemplate the information and use it to judge themselves and each other. This tactic is best explained by Foucault's perspective on surveillance strategies. He believes they are more about influencing an individual's psychology rather than trying to directly control what they do or make decisions for a person:

> Those who are subjected to a field of visibility, and who know it, assume responsibility for the constraints of power, it makes them play spontaneously upon themselves. They inscribe in themselves the power relation in which they play both roles. [31]

Operators can connect how their actions produce the data and can anticipate how their managers and coworkers will respond. They therefore preemptively discipline themselves, which is the intended effect of the management strategy [31]. As operators watch and modify themselves, they internalize the social norms of the managers and the organization as a whole who control them, effectively reproducing the same organizational 'spirit', work culture and other structures that are embodied in a spreadsheet's interpretive scheme, analytical model and normative accounting.

### 6.3. Management Control

The previous illustration shows just the basics of how a digital spreadsheet can apply simple logical functions coded into their design to automatically produce statistics that assess, compare and categorize people for the purpose of imposing social norms. It was demonstrated how the same information produced by underlying interpretive schemes embedded into spreadsheets that determine and validate the exercise of power through facilities also signify organizational expectations. These normative assessments in turn effectively transform spreadsheets into surveillance tools that anonymously and continuously evaluate the performance of workers in achieving certain intended organizational outcomes. Spreadsheet can have surveillance and disciplinary aspects that continuously inspect and measure workers and then use 'objective' performance outcomes to justify and validate management strategies aimed at regulating their conduct.

The consequence of reducing the conduct of workers to statistics is that they are then forced to respond to the public presentation of the information by self-correcting their behavior or at a minimum giving the superficial appearance of conformity. The statistics

represent a kind of 'shadow' of a worker that managers examine for the 'correct' outcomes. Winiecki uses the metaphor of a "shadowboxing screen" to describe the process where workers reflect the "organization's construction of them, back to themselves". Surveillance takes on the form of self-inspection or self-analysis. Using spreadsheets this way creates an apparatus of systematic, continuous and pervasive normalization, which eliminates the stress of getting caught doing anything 'wrong' because workers nearly always appear to be doing what is 'right'. The outcome of this interaction between worker and management mediated by information systems, according to Winiecki, is a "socially established 'objectivity' that is continuously influenced by its subjects as they are continuously influenced by it. The imputed 'objectivity' is actually a set of both ongoing programmatically compliant and agonistic actions with the apparatus" [32].

As workers do what managers expect and results improve, this validates decisions and the allocation of resources that are also facilitated by the same spreadsheets. Giddens emphasizes that the transformative capacity of organizations rests in this reflexive interaction between knowledge, agency and power. As Sharma et al. explains: "human agents continually monitor their actions and that of others. . .that in turn, reciprocates their decisions." [33]. Foucault refers to this phenomenon as a circular relation between 'truth' of the need for performance that defines what is 'right' and the power of disciplining practice through self-regulation: "Knowledge, once used to regulate the conduct of others, entails constraint, regulation and the disciplining of practice. Thus, there is no power relation without the correlative constitution of a field of knowledge, nor any knowledge that does not presuppose and constitute at the same time, power relations" [34].

## 7. Discussion of Results

A review found that the literature rarely precisely explores the particular ways information systems operate to reproduce the structures of social organizations. The present study addresses this gap by revealing how the familiar built-in properties (features, capabilities and analytical tools) of digital spreadsheets are appropriated by organizations to enact management techniques that enable or inhibit workers. It focuses on explaining how spreadsheets are easily reconstructed to contain not just information, but also organizational values, rules, norms, and other structures in their situated design that operate to conduct what people do in their situated use. It is shown through vignettes of spreadsheets-in-practice that digital spreadsheets are the tangible manifestations or objective forms of Giddens' three structural modalities (interpretive schemes, facility, and social norms) associated with various outcomes in the context of a social organization. Specifically, it explicitly shows how spreadsheet properties are appropriated to embed the aforementioned modalities.

In summary, the present article demonstrates how the underlying gridmatic framework (rows, columns and intersecting cells) of digital spreadsheets and their built-in validation rules are used to structure how data are maintained as constituted knowledge. This process produces an initial layer of meaning that makes spreadsheets powerful as archives of purposefully constructed and legitimated information. The visual tabular format of a spreadsheet also creates a geometric space that enables the use of complex formulas in the analysis of spreadsheet data. These formulate interpretive schemes that construct the intended meaning and significance of a dataset and frame it within the context of organizational objectives, thus predetermining how agents interact with the information. Interpretive schemes further enable the application of more complex analytical models that guide decision making and agents in exercising power by allocating the resources that spreadsheets track. Conditional functions based on Boolean logic overlay yet another level of meaning based on normative rules that are then used to judge the performance of workers and apply sanctions that induce workers into aligning their actions with organizational expectations.

The main result is a proposed framework (Figure 5) that illustrates the above process, which adapts and operationalizes Giddens' Structuration Theory for the analysis of digital

spreadsheets. It characterizes specific spreadsheet properties by their role in facilitating structural modalities and explains how spreadsheets offer workers interpretive schemes, facilities and social norms that influence their routine practices through their widespread diffusion and use in particular social organizational contexts. The model illustrates the sequential steps and work of data gathering, organization, analysis, decision making, and evaluation is a layered and unfolding process mediated by spreadsheets. The typology of spreadsheet properties that underly the structural modalities are viewed as three hierarchical overlapping and intertwined dimensions of meaning making. Together, they enable spreadsheets to produce 'effects of power' that manifest as the allocation of resources and the tangible routine practices of workers during their everyday work.

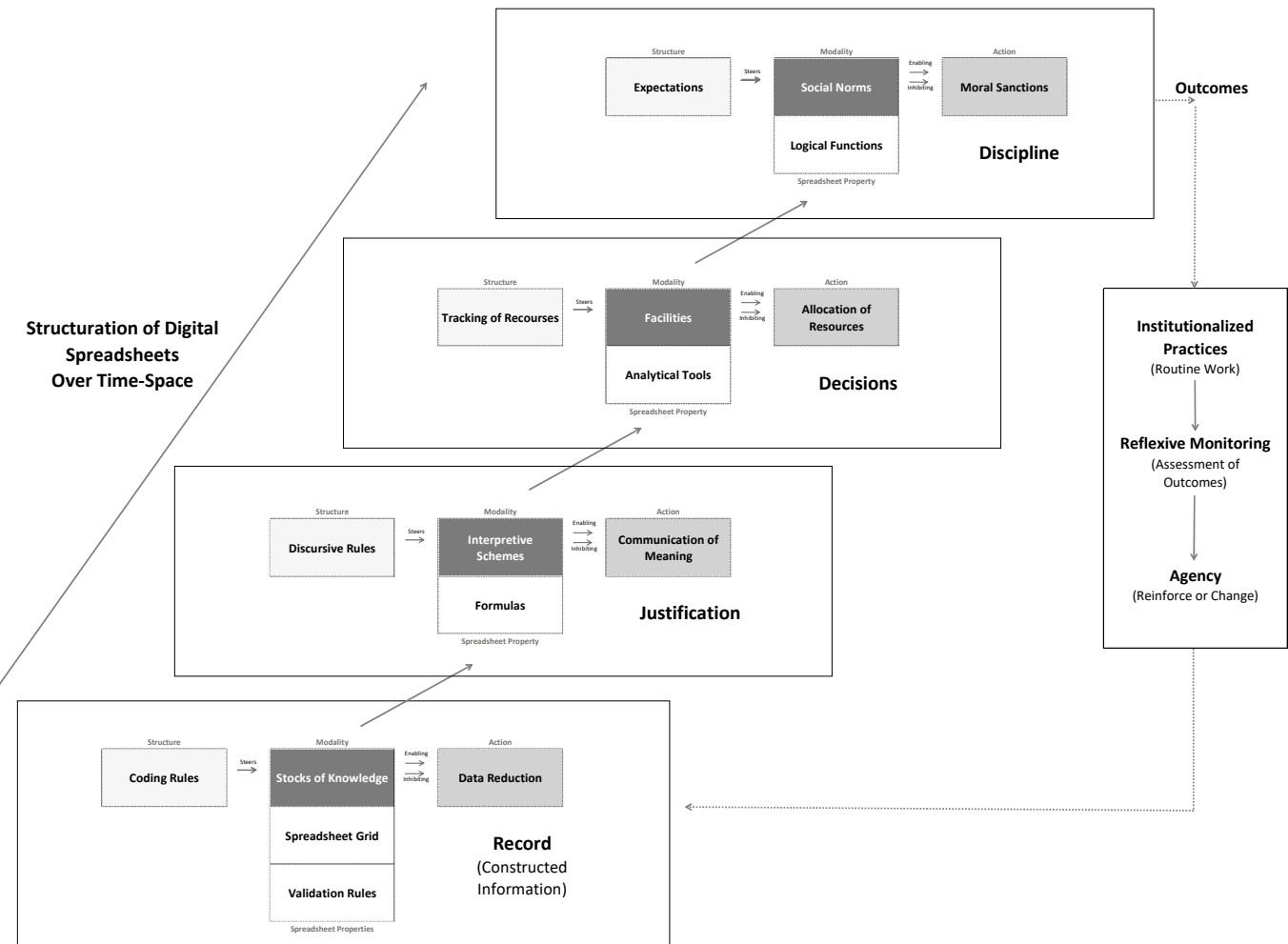

**Figure 5.** Proposed structurational stack framework for analyzing digital spreadsheets-in-Practice. Adapted from Giddens and Orlikowski [1,3].

This framework draws on Giddens' notion of the mutually constitutive duality to explain the emergence and change in spreadsheet technology through its use over time-space by highlighting the recursive evaluation of intended and unintended outcomes [1]. Drawing on Orlikowski, it offers a 'way of seeing' spreadsheets-in-practice as being continuously reciprocally shaped by—and resulting in—the persistent social practices that make up modern organizations [3]. The significance of the framework rests in the operationalization of structuration theory to characterize the specific spreadsheet properties that give material and visible forms to the abstract notions of structural modalities which are theorized to 'conduct the conduct' of knowledgeable human agents [27].

## 8. Implications

Giddens placed power, simply put, as the ability to get things done, and made it the central concept in Structuration Theory [35]. The present article argues that digital spreadsheets are 'instruments of power'. The appropriation of spreadsheet properties to embed structures are practical for facilitating organizational tasks such as information collection, decision making, allocating resources, tracking performance indicators, incentive schemes, and so on. However, these practices are rooted in a set of rules that reflect the diffusion of the values, intentions and truth claims of broader fields of power.

This poses a dilemma for the spreadsheet designers. In taking up the position that the main purpose of information systems is to support practical ways of understanding and using data to facilitate making the 'best' decisions and 'getting things done', this effectively sets into practice a definition of what counts as the 'right' mode of work and subjectively frames what it means to be 'good' worker. In turn, these expectations validate the construction of spreadsheets that are used to apply interpretive schemes to collect data that inspects, compares, judges, rates, and sorts workers for the purpose of assessing their 'performance' and actively disciplining them such that they become agents of their own subjectification under regimes of truth.

Giddens is most concerned about the negative consequences of disciplinary power and sanctions, which are linked to information collection and the development of surveillance, that are wielded by managers in organizations as means of control: "the generation of power presumes reflexively monitored system reproduction, involving the regularized gathering, storage and control of information applied to administrative ends . . . . and expressive of the sanctions that those in the apparatus are able to wield in respect of deviance" [20] (p. 178). When spreadsheets operate to collect information to detect when agents deviate from organizational norms, their trajectory then becomes intertwined with the purpose of identifying 'norm-violators' among those who are subject to their surveillance.

The power of spreadsheets is necessarily productive and more often than not their effects are viewed as benign; however, the present article shows how spreadsheets operate to privilege specific modes of interpretation and purposely produce knowledge to reinforce discourses. As was stated earlier, spreadsheets are not neutral, they are very much political. Their trajectory is often in the direction of becoming progressively productive through strategies of control and discipline. This represents an ethical concern for the information systems developers when they are conceivably directed to make changes to their systems for the purpose of reproducing the power relations in a social organization at the cost of actual system effectiveness and workers' autonomy.

The proposed framework of the present study has an application in illuminating the social consequences resulting from the trajectory of spreadsheets organizations employed to mediate peoples' work. Rose and Sheeper argue that the analysis of information systems usually narrowly concentrates on the "process, data, object, and entity of supposedly objectively observable business systems", rather than on the aspects explored in the present article involving "intended and actual social interactions" [7] (p. 228). Even though structuration theory may diverge from conventional information systems analysis, it is vitally important because of the ways spreadsheets, for example, make and remake reality to shape human subjectivity with insufficient organizational awareness of their conditions and consequences.

*Future Research Directions*

The continued focus of research in this area will be on how the structural dimensions of spreadsheets organize the properties of social systems through the interplay between the technology, workers and the organizational context within which they are used. The present study's insights call for further research to critique information systems, reflect on the roots of the realities that they produce, consider how they impact internal power relations in organizations, and uncover other ways they enable and restrict the actions of workers as they operate to stabilize structures across time and space. Specifically, future

research will explore the historical advancement of spreadsheet applications through the addition of new properties in response to the practices of management through spreadsheets. Future research would apply Giddens' notion of the recursive constitution of people and technology through structures to describe innovations and changes in digital spreadsheet applications during their diffusion over space and use over time.

**Author Contributions:** Conceptualization, P.A.R. and A.J.P.; methodology, P.A.R.; formal analysis, P.A.R. and A.J.P.; investigation, P.A.R. and A.J.P.; writing—original draft preparation, P.A.R. and A.J.P.; writing—review and editing, P.A.R. and A.J.P.; visualization, P.A.R. All authors have read and agreed to the published version of the manuscript.

**Funding:** This research received no external funding.

**Institutional Review Board Statement:** Not applicable.

**Informed Consent Statement:** Not applicable.

**Data Availability Statement:** Not applicable.

**Conflicts of Interest:** The authors declare no conflict of interest.

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
