# Peer review of "Knowledge, Decisions, and Norms: A Framework for Studying the Structuration of Spreadsheets in Social Organizations"

_information, doi:10.3390/info13020046_

Round 1

Reviewer 1 Report

An exciting, conceptually original text looks at how the tables are transcribed into the way organizations think and make decisions. I have almost no comments on the text in its partial aspects other than formal details:

lines 694 and 699 could be suitably separated when containing the formula.
- Figure 4 - needs to be delivered in better quality (higher resolution)

But I have two structural comments, and I leave it up to the authors to deal with them.

1. The theoretical part shows a gradual transition from a static structure to a specific dynamic understanding; every parameter, every behaviour of partial elements changes and influences the whole and at the same time, it is affected by it. While theoretical knowledge has arrived at these very dynamics, at the constant negotiation of equilibrium, research focuses on static tables and simple Boolean operations. Is this really an appropriate/adequate procedure?

2. Figure 5 and its interpretation (i.e. throughout the conclusion of the paper) gives a very rigid linear impression. This is indicated in the discussion. But the individual building blocks are very dynamic and interconnected (for example, displaying social norms as a "block" is problematic). Is there a way to maintain some appropriate description of the behaviour of companies and at the same time not to follow the path of clearly demarcated entities, which in most cases cannot be adequately defined? In my opinion, the problem is that if the authors properly defined all the terms in the scheme, they would find that it could not be constructed in this way. This is a strict approximation justified by an analysis of practice but not by theory.

Reviewer 2 Report

Dear authors, itwas a please to read your highly interesting manuscript in which you apply Gidden’s structuration theory tot he use of spradsheets in firms and their implications. It is very helpfult that you used case studies and examples to illustrate your insights.

The only recommendation I can give you is to improve the managerial implications section. Gidden’s theory is very abstract and, despite illustrating your thoughts, at the end of the article the reader may be left with the question how your findings can be used in practice. What should managers/leaders and employees/followers do (pay attention to…, question…, act…)?
